# Gut Microbiota-Derived Tryptophan Metabolites Maintain Gut and Systemic Homeostasis

**DOI:** 10.3390/cells11152296

**Published:** 2022-07-25

**Authors:** Xiaomin Su, Yunhuan Gao, Rongcun Yang

**Affiliations:** 1Translational Medicine Institute, Affiliated Tianjin Union Medical Center, Nankai University, Tianjin 300071, China; xiaominsu@nankai.edu.cn (X.S.); gaoyh_fei@163.com (Y.G.); 2State Key Laboratory of Medicinal Chemical Biology, Nankai University, Tianjin 300071, China; 3Department of Immunology, Nankai University School of Medicine, Nankai University, Tianjin 300071, China

**Keywords:** gut microbiota, Trp metabolites, anti-inflammatory macrophages, regulatory B cells, regulatory T cells

## Abstract

Tryptophan is an essential amino acid from dietary proteins. It can be metabolized into different metabolites in both the gut microbiota and tissue cells. Tryptophan metabolites such as indole-3-lactate (ILA), indole-3-acrylate (IAC), indole-3-propionate (IPA), indole-3-aldehyde (IAID), indoleacetic acid (IAA), indole-3-acetaldehyde and Kyn can be produced by intestinal microorganisms through direct Trp transformation and also, partly, the kynurenine (Kyn) pathway. These metabolites play a critical role in maintaining the homeostasis of the gut and systematic immunity and also potentially affect the occurrence and development of diseases such as inflammatory bowel diseases, tumors, obesity and metabolic syndrome, diseases in the nervous system, infectious diseases, vascular inflammation and cardiovascular diseases and hepatic fibrosis. They can not only promote the differentiation and function of anti-inflammatory macrophages, Treg cells, CD4^+^CD8αα^+^ regulatory cells, IL-10^+^ and/or IL-35^+^B regulatory cells but also IL-22-producing innate lymphoid cells 3 (ILC3), which are involved in maintaining the gut mucosal homeostasis. These findings have important consequences in the immunotherapy against tumor and other immune-associated diseases. We will summarize here the recent advances in understanding the generation and regulation of tryptophan metabolites in the gut microbiota, the role of gut microbiota-derived tryptophan metabolites in different immune cells, the occurrence and development of diseases and immunotherapy against immune-associated diseases.

## 1. Introduction

Tryptophan (Trp) is an essential amino acid, which can be naturally provided by dietary proteins such as poultry, milk, tuna, fish, cheese, bread, oats, prunes, chocolate and peanuts. It is a biosynthetic precursor of a large number of metabolites [1]. Trp metabolites can be derived from the direct transformation of Trp by intestinal microorganisms, such as indole, tryptamine, indole ethanol (IE), indolepropionic acid (IPA), indolelactic acid (ILA), indoleacetic acid (IAA), skatole, indolealdehyde (IAld) and indoleacrylic acid (IA) [2,3,4,5,6], and from the Kyn pathway, such as Kyn and 3-hydroxyanthranilic acid (3-HAA) [1,7]. These metabolites play a critical role in maintaining the epithelial cell structure and function [8], gastrointestinal motility [9] and insulin secretion of pancreatic β cells [10]. These Trp metabolites can also maintain the homeostasis and function of immune cells [11,12]. They activate signal pathways to control the differentiation and functions of immune cells through transcription factors such as the aryl hydrocarbon receptor (AhR), which is expressed in macrophages (Macs) [13], dendritic cells (DCs) [14], T cells [15], B cells [16], CD4^+^CD8αα^+^ lymphocytes [17] and innate lymphoid cells (ILCs) [18]. Trp metabolites also induce IL-22 production in T cells and promote the differentiation of IL-22-producing ILC3 cells.

Gut microbiota-derived Trp metabolites are also related to many diseases, such as inflammatory bowel diseases, vascular inflammation and cardiovascular diseases, hepatic fibrosis, diseases in the nervous system, obesity and metabolic syndrome, infectious diseases and tumors. The recent progress in large-scale sequencing and mass spectrometry has further increased our understanding of the influence of the gut microbiota and its metabolites on the progression of tumors and the efficacy of immunotherapy on tumors. Their effects on stress-related depression, schizophrenia and Alzheimer’s and Parkinson’s diseases are widely known [19]. There are significant differences in some Trp metabolites and metabolic enzymes between patients and healthy volunteers, such as that patients with inflammatory bowel diseases (IBD) have lower levels of Trp in both their serum and feces than healthy subjects [20,21]. Trp metabolite Kyn is present in higher amounts in the plasma of advanced-stage cancer patients. Interestingly, Trp metabolites and their interacting ligands are also involved in tumor immunotherapy. A high-serum Kyn/Trp ratio is also correlated with a poor prognosis after a PD-1 blockade in lung cancer, melanoma and renal cell carcinomas [22]. We summarize the recent advances in understanding the generation and regulation of Trp metabolites in gut microbiota, the regulation of gut microbiota-derived Trp metabolites in the different immune cells and potential applications in immunotherapy against immune-associated diseases such as tumors.

## 2. Trp Metabolism in Gut Microbiota

### 2.1. Trp Metabolism in Gut Microbiota

Trp metabolism in the gut microbiota has been reviewed by multiple papers [1,23,24,25]. The degradation of dietary proteins leads to the release of Trp, which is converted into various metabolites by the gut microbiota, such as indole, indole-3-acid-acetic (IAA), indole-3-propionic acid (IPA), indoleacrylic acid (IA), indole-3-aldehyde (IAld), tryptamine, indoleethanol (IE), indole-3-acetaldehyde (IAAld) and 3-methylindole (skatole), or catalyzed to produce Kyn and downstream metabolites such as 3-hydroxyanthranilic acid (3-HAA) and 3-hydroxykynurenine (3H-Kyn) [1,23,24] (Figure 1 and Table 1). Bacterial species possess different catalytic enzymes, whereas some bacteria can also cooperate with each other to generate Trp metabolites. Bacterial species such as *Bacteroides ovatus*, *Clostridium limosum*, *Enterococcus faecalis* and *Escherichia coli* are able to convert Trp into indole [23,26]. The oxidative and reductive pathways in *Clostridium sporogenes* lead to the production of IAA and IPA [20,27]. *Clostridium bartlettii* and *Bifidobacterium* spp. produce ILA and IAA [28]. *Peptostreptococcus* spp. convert Trp to IA and IPA [29]. IAld is also generated by *Firmicutes phylum*, such as *Lactobacillus (L). reuteri*, *L. *johnsonii**, *L. acidophilus* and L. murinus, via the aromatic amino acid aminotransferase (ArAT) and an indolelactic acid dehydrogenase (ILDH) [17]. Skatoles are generated by the decarboxylation of IAA by *Bacteroides* spp. and *Clostridium* spp. [5,28]. *Ruminococcus gnavus* converts Trp into tryptamine by a Trp decarboxylase enzyme [30]. Trp-metabolizing pathways also exist in some members of the human gut microbiota, such as *Clostridium sporogenes*, which can decarboxylate Trp to the neurotransmitter tryptamine [30].

In addition, several intestinal bacteria encode enzymes homologous to those of the eukaryotic Kyn pathway. These enzymes can produce Kyn and downstream metabolites such as 3-hydroxyanthranilic acid [7]. However, it will be necessary for understanding which kind of gut microbiota produce Trp metabolites [24], more bacterial strains with a catalytic capacity against Trp remain to be identified. This will be beneficial for the design of targeted strategies to control Trp metabolite production.

### 2.2. Regulation of Trp Metabolism in Gut Microbiota

Recent studies showed that Trp metabolism in the gut microbiota can be regulated by multiple factors such as natural substances and chemical drugs (Figure 1). Ginsenoside Rg1 could increase the levels of Trp metabolites in the serum, including indole-3-carboxaldehyde, indole-3-lactic acid, 3-indolepropionic acid and niacinamide [43]. In mice with high-fat diet-induced obesity, ginsenoside Rb1 significantly altered the gut microbiota composition and serum Trp [44]. Fucose increased the abundance of Trp-producing *E. coli* and normalized the blood Trp levels [45]. Fructooligosaccharides (FOS) modulated the gut microbiome profiles, such as an increase in the abundance of *Ruminococcacere* (phylum level) and a decrease in the abundance of *Akkermansiaceae* (family level) and *Verrucomicrobia* (phylum level), and significantly increased the levels of Trp and 5-hydroxytryptamine (5-HT) [46]. The ginseng polysaccharides decreased L-kynurenine, as well as the ratio of Kyn/Trp [47]. The Fuzhuan brick tea polysaccharide contributed to the proliferation of beneficial microbiota, such as *Lactobacillus* and *Akkermansia*, and altered the Trp metabolism and elevated the fecal contents of IAld and IAA [48]. *Flammulina velutipes* polysaccharides (FVPs) also changed the composition of the gut microbiota, which affected the Trp metabolism [49]. The low abundance of *Escherichia*-*Shigella*, *Dubosiella* and *Allobaculum*, along with the enrichment of *Muribaculaceae_unclassified*, *Ralstonia* and *Rikenellaceae_RC9_gut_group* in the gut, which could result in higher Trp metabolite levels, could be detected in yellow wine polyphenolic compound-treated rats [50]. Shenling baizhu san also altered the gut microbiota structures and increased the microbial levels of the Trp metabolites, including indole-3-propionic acid and indole-3-acetic acid [51]. *Myristica fragrans* regulated the gut microbes and metabolites to activate Trp metabolite-mediated AhR in mice fed a high-fat diet [52]. Pu-erh tea could boost the indole and 5-hydroxytryptamine pathways of the Trp metabolism [53]. The metabolomic analysis revealed metabolic profile alternations in response to the gut microbiota reprogrammed by a qingchang wenzhong decoction (QCWZD), especially enhanced Trp metabolism [54]. Gallnut tannic acid and the zingiber officinale–panax ginseng herb pair also stimulated the growth of the beneficial bacteria and suppressed the growth of the pathogenic bacteria [55,56].

In addition, studies also found that some chemical drugs also modulate Trp metabolism in gut microbiota, such that Fisetin can modulate the gut microbiota-mediated Trp metabolism [57]. Rifaximin affected Trp synthesis in the gut microbiota [58]. Diallyl disulfide (DADS) altered the gut microbial community structure and metabolic profile in mice [59].

## 3. Regulation of Trp Metabolites from Gut Microbiota in the Immune Cells

Trp metabolites play an important role in the differentiation and function of T-regulatory cells (Tregs), B-regulatory cells (Bregs), IL-22-producing innate lymphocyte cells 3 (ILC3) and anti-inflammatory macrophages (Figure 2). Trp metabolite receptors such as the ary hydrocarbon receptor (AhR) can be detected in T cells such as T-helper type 17 (Th17) cells and Tregs [60], B cells and antigen-presenting cells (APCs) [61]. When bound to its ligand, AhR is located in the cytosol of the cells and translocated into the nucleus to heterodimerize with the AhR nuclear translocator and to target the gene promoter. For AhR, the most effective Trp metabolites are indole, skatole, IA, tryptamine, IPyA and indole-3-acetamide (IAM), whereas IAA, IAID, IPA and ILA are the least active [62,63].

### 3.1. Macrophages

(1)Inflammatory and immunosuppressive macrophages. In vitro studies show that Trp metabolites suppress inflammatory responses in macrophages. Trp metabolite receptor AhR signaling has an important role in the function of macrophages [64]. AhR is required for the amelioration of *Streptococcus* and *Salmonella typhimurium*-induced immunopathology in LPS-tolerant mice [65]. AhR knockout (KO) mice were more sensitive to LPS-induced lethal shock [66]. These AhR KO mice can produce higher amounts of proinflammatory cytokines (TNF, IL-6 and IL-12). Others also found that the decreased inflammatory processes in LPS-activated macrophages by an endogenous (FICZ) or exogenous (BaP) ligand is partially dependent on AhR signaling [67]. AhR downregulates the production of the proinflammatory cytokine IL-6 through suppressing histamine production in macrophages [68]. The immunomodulatory roles of AhR are through a Rac1 ubiquitination-dependent mechanism, which can attenuate AKT signaling and result in a mitigated inflammatory response [67]. The proteomic changes in the macrophages after treatment with Trp metabolites I3AA or IAld, as well as AhR ligand benzo(a) pyrene (BaP), showed that fatty acid β-oxidation and oxidative phosphorylation were significantly increased in a time- and LPS-dependent manner [69]. In addition, the AhR-Src-STAT3-IL-10 signaling pathway is also a critical pathway in regulating inflammatory macrophages [70].

Kyn from Kyn pathway-mediated immunosuppression depends on the interplay between tumor-associated macrophages and Tregs in tumor environments [71]. Kyn also interacts with ligand-activated AhR to drive the generation of tolerogenic myeloid cells. Studies have shown that Kyn can reach concentrations sufficient to activate the AhR pathway in some tumor microenvironments [72]. Kyn downstream metabolite 3-HAA inhibits the PI3K/Akt/mTOR and NF-κB signaling pathways and decreases the production of proinflammatory cytokines, IL-6 and TNF-α in macrophages [73]. Thus, Trp metabolites from Trp transformation and the Kyn pathway by the gut microbiota suppress inflammatory responses in the macrophages.

(2)Gut resident macrophages. It is unclear how Trp metabolites regulate these resident macrophages. In gut tissues, Muller et al. [74] discussed the origin, phenotype and function of the resident macrophages in the different layers of the intestines during homeostasis. A “monocyte waterfall” from circulation to the intestine maintains the macrophage pool in a CCR2-dependent manner in the murine colon [75]. The monocytes, which are identified as the ly6c^hi^ CX3CR1^int^ MHCII^−^ subset, exhibit proinflammatory properties. They terminally differentiate into mature resident ly6c^low/−^CX3CR1^hi^MHCII^hi^ macrophages in the intestines. The resident macrophages reside in the lamina propria (LP) or the muscle layer. LP macrophages (LPMs) may be subdivided into mucosal and submucosal LPMs [76], whereas mucosal LPMs line the intestinal epithelium and vasculature in the intestines [77,78]. These macrophages contribute to the host defense, barrier integrity and constitutive secretion of interleukin (IL)-10, which can promote the maintenance of FoxP3^+^ Treg [79]. Perivascular macrophages participate in the regulation of the vasculature in the small intestine and colon [76,78]. The macrophages residing in the muscularis are essential for tissue homeostasis. Their interactions in muscularis macrophages with the neurons control intestinal motility and protect tissues. These macrophage subsets in gut tissues belong to resident macrophages (anti-inflammatory macrophages). However, how Trp metabolites regulate these resident macrophages remains unclear. (3)Myeloid precursor cells. The activation of Trp metabolites receptor AhR inhibits the proliferation of myeloid precursor cells [80]. In vitro studies showed that AhR could influence the monocyte fate to drive the differentiation of DCs over macrophages, suggesting that AhR plays a key role in the macrophage–dendritic cell balance in inflamed tissues [64]. AhR also inhibits human CD34^+^ hematopoietic precursor cells from differentiating into monocytes and Langerhans cells (LCs) [81].

### 3.2. Regulatory T Lymphocytes

(1)Regulatory CD4^+^Foxp3^+^T cells. T-regulatory cells include multiple subsets such as CD4^+^Foxp3^+^ (Tregs), CD4^+^Foxp3^−^IL-10^+^(Tr1s) and IL-10^+^CD8 regulatory cells. Tregs express transcription factor Foxp3 [82,83] and differentiate in the thymus or the periphery [84]. They constitutively express inhibitory molecules such as cytotoxic CTLA-4 [85]. These Tregs play a critical role in suppressing tissue inflammation through the release of cytokines such as TGFβ, IL-10 and IL-35. Gut tissue-resident Tregs can be induced in response to dietary and gut microbiota [86,87]. Studies have shown that AhR is important in Tregs by controlling the production of IL-10 and IL-22 [88,89,90,91,92]. Indeed, the immunosuppressive effect of TCDD, an AhR ligand, is linked to the expansion or induction of Treg cells and promotion of the function of Tregs in mice and in humans [89,90]. Another ligand of AhR (2-(1′H-indole-3′-carbonyl)-thiazole-4-carboxylic acid methyl ester (ITE)) also suppresses autoimmunity by inducing Tregs [93]. The activation of AhR with ITE can suppress IBD [94] and ameliorates encephalomyelitis (EAE) symptoms [95]. 4-n-nonylphenol, an agonist for AhR, can induce Tregs [96]. Indole and its derivatives from Trp regulate Tregs through the AhR–ligand–Treg axis, thereby affecting the function of Tregs [97,98]. Notably, *AhR* expression in the Tregs of the spleen and lymph nodes is very low, whereas *AhR* is highly expressed in intestinal Tregs [99].

In addition, Kyn in the Kyn pathway by gut microbiota can enhance Treg cell differentiation through the activation of AhR [65,72]. It induces the differentiation of naive CD4^+^ T cells into immunosuppressive FoxP3^+^ Tregs and not proinflammatory Th17 cells [100]. Kyn metabolites also increase FoxP3^+^ Tregs through direct transactivation and the induction of epigenetic modifications that control Foxp3 transcription and, also, through the modulation of DCs [100,101,102]. 3-HAA, a downstream metabolite of Kyn, increases the generation of Foxp3^+^ T_reg_ cells and immunosuppressive TGF-β in a nuclear coactivator 7-dependent pathway [103].

(2)Regulatory type 1 (Tr1) cells. Regulatory type 1 (Tr1) cells are an important subset of CD4^+^ T cells in the control of excessive inflammatory responses [104]. These cells are characterized by IL-10 expression but not Foxp3. Tr1 cells are prominent in chronic infections and immune manipulations in vivo [105]. They are described with regulating activity due to their tolerance to foreign antigens and their capacity to inhibit the proliferation of lymphocytes [106]. The activation of Trp metabolites receptor AhR supports the differentiation of type 1 regulatory T cells (Tr1) [92] and also promotes the differentiation of CD4^+^Foxp3^+^ T cells, which can produce IL-10, and control responder T cells [89]. During Tr1-cell differentiation, AhR physically associates with c-Maf and transactivates the IL-10 and IL-21 promoters [88]. AhR also promotes HIF1-α degradation and takes control of Tr1 cell metabolism [92]. In addition, AhR activation can also initiate the differentiation of mucosal-homing Tim3^+^Lag3^+^Tr1 cells [107]. Thus, Trp metabolites promote the differentiation and function of not only Treg cells but also other regulatory T cells, such as Tr1 cells.

### 3.3. Regulatory B Cells

Regulatory B lymphocytes (Bregs) have been described for decades in mice and in humans. These Bregs correspond to diverse subpopulations of B cells according to their phenotypes and/or activities. In mice, multiple kinds of Breg cells have been identified, including CD5^+^CD1d^high^ B10 cells, CD5^+^ B1a cells, CD21^hi^CD23^hi^CD24^hi^CD1d^hi^ transitional type 2 MZ precursor (T2-MZP) cells, CD21^high^CD23^−^MZ cells, Tim1^+^ B2 cells, CD138^+^CD44^high^ plasmablast, CD24^high^CD27^+^ memory cells and CD138^+^B200^+^ plasma cells [108]. They express different markers in these Bregs, including IgD, IgM, IL-10, CD1d, CD5, CD11, CD21/CD35, CD23, CD24, CD25, CD69, CD72, CD138, CD40 and CD 86 [109,110,111]. The population of increased IL-35^+^ Bregs in *huREG4^IECtg^* mice has been identified as

CD19^+^IgM^+^IgD^+^IL10^+^CD1d^high^CD5^low^CD11b^low^CD21/CD35^Low^CD23^Low^CD25^Low^ CD72^low^CD69^−^CD138^low/−^CD40^low/−^CD86^low/−^ cells [112], which are different from IgG-producing Bregs but similar to IgM^+^IgD^+^ Bregs [110]. These Bregs restrain excessive inflammatory responses [113] and contribute to the maintenance of immunological tolerance [114]. They exert wide effects on multiple types of immune cells, such as T cells, macrophages, DCs and B cells [115]. They promote the generation of Tregs and anti-inflammatory macrophage (M2) [116,117] and impede the differentiation of Th1 cells [118]. Bregs suppress a variety of immune pathologies, including autoimmune diseases, through the production of interleukin (IL)-10, IL-35 and TGFβ1 [114]. These Bregs also play a critical role in regulating immunity in multiple diseases, such as cancer progression and autoimmune and infectious diseases [16,109,115,119,120]. Trp metabolite receptor AhR participates in B-cell differentiation, maturation and activation [121,122]. The differentiation and function of IL-10-producing CD19^+^CD21^hi^CD24^hi^ Bregs can be regulated by AhR [16]. The relationship between IL-35 and NFκB has been reported. IAA by the gut microbiota, together with LPS, can activate NF-κB through TLR4 to induce the generation of IL-35^+^ cells [112]. IAA with LPS promotes the activity of transcription factors PXR, RXR and CAR, which are necessary for the expression of IL-35 [112]. B cells isolated from WT mice also increase the expression of p35 and Ebi3 upon activation via TLR4 [109]. Influenza A virus (IAV)-mediated IL-35 is regulated by NF-κB [123]. Direct communication between intestinal symbionts and PXR regulates the mucosal integrity through luminal sensing and signaling by TLR4 [124]. Thus, Trp metabolites can induce regulatory IL-10^+^ and /or IL-35^+^ B cells.

### 3.4. IL-22-Producing Cells

The cells such as αβ T cells, γδ T cells, natural killer T cells (NKT cells) and ILCs can produce IL-22 [125]. ILCs, a heterogeneous lymphoid cell subset, lack T-cell and B-cell antigen receptors [126]. These cells are early responders in the initiation of inflammatory responses, which make these ILCs a significant subject in the field of AhR immunity [127]. In three different groups of ILCs, namely, ILC1s, ILC2s and ILC3s, only ILC3s are IL-22 producers [126]. IL-22 is crucial for the maintenance of IECs and the defense against pathogens [128]. Trp metabolite receptor AhR plays a central node in ILC3 development. *L. reuteri* uses Trp to expand and generate an AhR ligand IAld, which contributes to IL-22 transcription in innate lymphoid cells and T cells [129]. AhR ligands by microbiota (such as IAld generated by *Lactobacilli*) foster IL-22 production by ILC3s [34]. AhR is an important transcription factor for all ILC3 subsets, such as lymphoid tissue-inducer (LTi)-like ILC3s and NKp46^+^ ILC3s [130,131]. It is required for IL-22^+^ ILC3s in the first 2 to 3 weeks after birth, which is likely acquired from maternal AhR ligands [132]. Runx3 and its downstream target RORγT-mediated ILC3 development is through the induction of AhR [133]. Ikaros prevents ILC3 development through the inhibition of AhR expression [134]. Studies using AhR-deficient mice show that impaired AhR activity is correlated with diminished levels of IL-22–producing ILC type 3 (ILC3) and worsening inflammatory diseases [20]. Although the molecular mechanism underlying the development and function of ILC3s regulated by AhR is incompletely understood, AhR can increase ILC3 survival by the IL-7/IL-7R pathway and antiapoptotic gene expression [131]. Following activation, AhR can also induce and stabilize the expression of Notch [130] and c-Kit, which are necessary for ILC3 development [135]. In addition, decreased AhR signaling in ILC3s can alter the balance between ILC3 and ILC1 populations and promote the development of colitis. ILC2s also express high levels of *AhR*, which is intestine-specific and mediated by cooperative action between AhR and growth factor-independent 1 (GFI1) [136]. *AhR* also drives the expression of IL-22 in Th17 cells but is not required for their differentiation [60]. Thus, Trp metabolites induce IL-22-producing cells.

### 3.5. CD4^+^CD8αα^+^ Lymphocytes

There is a unique population of CD4^+^CD8αα^+^ TCRαβ T cells in the small intestinal epithelium [86]. These cells are found in mice colonized with *Lactobacillus reuteri (L. reuteri)*. *L. reuteri* can induce CD4^+^CD8αα^+^ IELs in germ-free (GF) mice, but conventionally raised mice lack these cells [17]. *L. reuteri*, which expresses high levels of aromatic aminotransferase, converts Trp into AhR activators and drives the reprogramming of CD4^+^ T cells into CD4^+^CD8αα^+^ IELs in the gut [17]. These cells have a regulatory function, which can be complementary to that of Tregs and promote tolerance to dietary antigens [92].. This population of cells requires AhR not only for their survival but also for their generation [17]. After AhR activation after exposure to Ficz, CD4^+^CD8αα^+^ IELs resisted apoptosis and upregulate IL-15 and IL-10 in a colitis model [137].

### 3.6. Other Immune Cells

Trp metabolite Kyn can interact with the ligand-activated transcription factor AhR to upregulate the expression of PD-1 in CD8^+^ T cells [71]. Kyn has been shown to be a ligand for GPR35, which is expressed in human CD14^+^ monocytes, T cells, invariant NKT (iNKT) cells, DCs, neutrophils, eosinophils and basophils [138]. Kyn metabolites, including Kyn itself, also suppress natural killer cells (NK) [139] and antigen-presenting cells (APC), such as DCs, monocytes and macrophages in mice [140,141]. 3-HAA metabolites of Kyn also cause immune suppression by inducing T-cell apoptosis through glutathione depletion [140]. In addition, Trp metabolites can suppress proinflammatory Th1 and Th17 [142].

## 4. Trp Metabolites from Gut Microbiota and Immune-Associated Diseases

Trp metabolites enhance intestinal epithelial barrier functions by increasing the expression of genes, which are involved in the maintenance of the epithelial cell structure and function [8]. The metabolite IPA regulates the intestinal barrier function in mice by activating the pregnane X receptor (PXR) [124] or AhR [29]. Indole modulates the secretion of glucagon-like peptide-1 (GLP-1) in mouse colonic enteroendocrine L cells [143], which is critical in stimulating the insulin secretion of pancreatic β cells, suppressing appetite and slowing gastric emptying [10]. Tryptamine induces the release of the neurotransmitter 5-HT, a serotonin of enterochromaffin cells [9]. 5-HT can work on the enteric nervous system to stimulate gastrointestinal motility [9] such as irritable bowel syndrome (IBS) [144]. However, Trp metabolites generated by the gut microbiota also contribute to intestinal and systemic homeostasis through regulating immune cells. These metabolites, such as tryptamine, skatole, IAA, IA, IAld and ILA, can affect immune responses through AhR [17,34]. We mainly summarize the regulation of Trp metabolites from the gut microbiota in immune-associated diseases (Figure 2).

### 4.1. Inflammatory Bowel Diseases

Some Trp metabolites and metabolic enzymes are significantly different in healthy individuals and patients with IBD. These patients have lower levels of Trp in the serum and feces than healthy subjects [20,21]. Trp metabolite IPA in the serum from patients with active colitis is also selectively diminished [145]. IBD patients have reduced fecal concentrations of the AhR agonist IAA [20]. Notably, others also found increased Kyn or Kyn/Trp ratios in IBD patients, indicating the promoted Trp metabolism along the Kyn pathway in active IBD [146]. In addition, a Trp-free diet also increases the susceptibility to DSS-induced inflammation in mice [147]. These observations suggest that the changes in Trp metabolism are involved in the etiology of IBD. Trp metabolites can modulate IBD by affecting the immune system [148]. Indole metabolites and kynurenine interact with AhR to induce T-regulatory cell differentiation, confine Th17 and the Th1 response and produce anti-inflammatory mediators.

### 4.2. Tumors

Carcinogenesis is interrelated with the human immune status and environmental factors; among which, the gut microbiota and its metabolites have been discussed widely over the past decade. The microbiota are now identified as an enabling factor in the most recent iteration of the ‘hallmarks’ of cancer [149]. The evidence shows that bacterial Trp metabolites play a role in the development of different types of cancer [72,150], such as bacterial indoles, which play an important role in colon carcinogenesis [151]. The gut microbiota activates AhR through the Trp metabolite kyn to mediate renal cell carcinoma metastasis [152]. However, the microorganisms within the gastrointestinal tract can also shape the overall immunity and influence the states of health and disease (including cancer) at the systemic level [153]. Trp catabolism is also reported to play immunosuppressive actions across many types of cancer [154]. Hezaveh et al. [155] recently revealed that metabolites of dietary Trp generated by the gut microbiota activate the aryl hydrocarbon receptor in myeloid cells, promoting an immune suppressive tumor microenvironment and facilitating pancreatic ductal adenocarcinoma growth.

### 4.3. Obesity and Metabolic Syndrome

Metabolites and bacterial components of the gut microbiota affect the initiation and progression of type 2 diabetes (T2D) and the metabolic syndrome by regulating inflammation, immunity and the metabolism. A correlation between the Kyn/Trp ratio and obesity, as well as the metabolic syndrome, has been reported [156]. The bacterially derived Trp metabolites indoles, IPA and indole sulfuric acid (ISA) are lower in blood samples from subjects with type 2 diabetes as compared to the lean controls. Higher serum concentrations of IPA are also associated with a reduced prevalence of T2D [157]. Recent studies have found that a higher milk intake and higher fiber intake were associated with a favorable profile of circulating Trp metabolites for T2D [158]. Several indole derivatives produced via Trp transformation by the gut microbiota have a role in metabolic syndrome pathogenesis, such as IAA-mediated IL-35^+^ Breg cells, which can affect high-fat diet-mediated obesity [112].

### 4.4. Diseases in Nervous System

Trp metabolism is also related to disorders of the nervous system. The gut microbiota can be involved in neuropsychiatric disorders. Its effects on stress-related depression, schizophrenia and Alzheimer’s and Parkinson’s diseases are comprehensively reviewed [19]. Trp metabolites of the gut microbiota have an effect on astrocytes to limit nervous system inflammation [159]. There are decreased circulating levels of AhR agonists in individuals with multiple sclerosis [159]. The plasma levels of IPA are significantly lower in subjects with Huntington’s disease compared to healthy controls [160]. 

### 4.5. Infectious Diseases

Studies show that Trp metabolites are also related to infection diseases. Some bacteria, such as *Mycobacterium tuberculosis*, escape the CD4-mediated defense by synthesizing their own Trp under stress conditions [161]. The degradation of AhR ligands leads to increased susceptibility to *Citrobacter rodentium* infection [131,162]. Trp metabolites promote the production of IL-22, which is a key cytokine in colonization resistance against fungi [34]. 

### 4.6. Vascular Inflammation and Cardiovascular Diseases

Recent studies indicate that indoles activate AhR and PXR receptors to affect the immune system’s function and further promote human health, including vascular regulation [163]. Indoxyl sulfate could promote vascular inflammation, whereas indole-3-propionic acid and indole-3-aldehyde had protective roles. Increasing evidence shows the protective role of microbiota-derived indole derivatives in blood pressure regulation and hypertension [164]. 

### 4.7. Hepatic Fibrosis

The gut microbiome influences liver diseases. Interactions between Trp metabolism, the gut microbiome and the immune system can be potential drivers of non-alcoholic fatty liver disease [165]. Indole-3-propionic acid, a gut-derived Trp metabolite, is associated with hepatic fibrosis [166]. Circulating IPA levels were found to be lower in individuals with liver fibrosis compared to those without fibrosis [166]. 

## 5. Potential Application of Gut Microbiota-Derived Trp Metabolites in Immunotherapy

Since Trp metabolites possess wildly regulatory functions in the gut and systemic immune system, Trp metabolites such as Kyn may be important targets in immuno-therapy against diseases in the nervous system, inflammatory bowel diseases, obesity and metabolic syndromes, atherosclerosis and tumors, with particularly potential implications in tumor immunotherapy, such as those related to checkpoint blockade immune intervention strategies [167].

Mice fed a Trp-supplemented diet had reduced inflammation and decreased severity of dextran sodium sulfate (DSS-)-induced colitis [168]. The blockade of the kyn–AhR axis can ameliorate colitis-associated colon cancer through inhibiting the immune tolerance [169]. Multiple natural substances such as ginsenoside Rg1 [43], fucose [45], fuzhuan brick tea polysaccharide [48] and shenling baizhu san [51] could alleviate ulcerative colitis by modulating the gut microbiota and microbial Trp metabolism.

Gut microbiota are the sources of additional Trp metabolites that affect antitumor immunity. Dietary and bacterial indoles have shown promising therapeutic targets for carcinogenesis [2]. Kyn depletion with a PEGylated Kyn-degrading enzyme causes cancer control in preclinical settings [170]. Trp metabolites have also produced profound impacts in the patients with specific tumors during immune checkpoint blockade therapy [171]. A higher Kyn/Trp ratio is correlated with a poor prognosis after a PD-1 blockade in lung cancer, melanoma and renal cell carcinomas [22]. Ginseng polysaccharides can alter the gut microbiota and Kyn/Trp ratio, which promotes the immunotherapy of anti-programmed cell death 1/programmed cell death ligand 1 (anti-PD-1/PD-L1) [47]. The gut microbiota regulates white adipose tissue inflammation and obesity via a family of Trp-derived metabolite-associated miRNA [172]. The potential contribution of Trp metabolites has been found in heart failure [173]. Clinical evidence also supports that metabolites of the kyn pathway are used as clinical biomarkers in various manifestations of coronary artery disease [174]. Fructooligosaccharides protect against OVA-induced food allergy in mice by regulating the Th17/Treg cell balance using Trp metabolites [46]. Fisetin could improve hyperuricemia-induced chronic kidney disease via regulating the gut microbiota-mediated Trp metabolism [57]. Interestingly, honeybee gut lactobacillus modulates host learning and memory behaviors via regulating the Trp metabolism [175].

## 6. Conclusions

Trp metabolites can not only promote the differentiation and function of anti-inflammatory macrophages, Treg cells, CD4^+^CD8αα^+^ immune regulatory cells, IL-10^+^ and/or IL-35^+^B regulatory cells but also IL-22-producing ILC3, which are involved in maintaining the gut and systemic homeostasis. Although some bacteria are capable of producing Trp metabolites, the contributors in the human gut remain, to a large extent, unknown. Since Trp metabolites are also produced by other microorganisms, it is also imperative to move away from profiling only the bacterial community. Once we identify the relevant microorganisms and Trp metabolites, we should also identify the exact role of each Trp metabolite in the host path physiology and unravel their precise mechanisms in the different cells of the intestines and other tissues.

Trp metabolites play a role in neurological, metabolic and psychiatric diseases; intestinal disorders and tumors. AhR and its interacting ligands are involved in tumor immunotherapy. Based on the immunosuppressive and cancer-promoting effects of Trp metabolites, Trp degradation remains an important target in immuno-oncology. However, the in-depth molecular mechanisms remain as yet unclear. In addition to finding more factors that can regulate Trp metabolism in the gut microbiota, next-generation probiotics, which can produce Trp metabolites, will be mostly identified between healthy and unhealthy individuals [176]. Those also include recombinant microorganisms, which can overexpress the genes of interest. This could represent an excellent alternative approach to modulate the host physiology.

## Figures and Tables

**Figure 1 cells-11-02296-f001:**
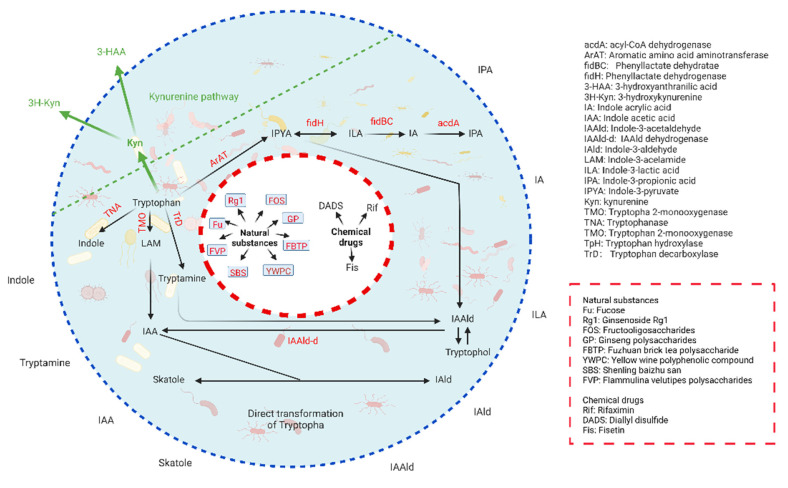
Production and regulation of Trp metabolites in the gut microbiota. Trp is an essential amino acid, which can be naturally provided by dietary proteins. Trp metabolism in the gut microbiota follows two major pathways: (1) Direct transformation. Trp is converted into various catabolites by the gut microbiota, such as indole, IAA, IPA, IA, IAld, tryptamine, IE, IAAld and Skatole. (2) Kynurenine pathway. Trp is catalyzed to produce Kyn and downstream metabolites such as 3-HAA and 3H-Kyn. There are some factors, including natural substances and chemical drugs inside the circle (red dotted line), which can regulate tryptophan metabolism in the gut microbiota. Words in red are a key enzyme. Background, gut microbiota.

**Figure 2 cells-11-02296-f002:**
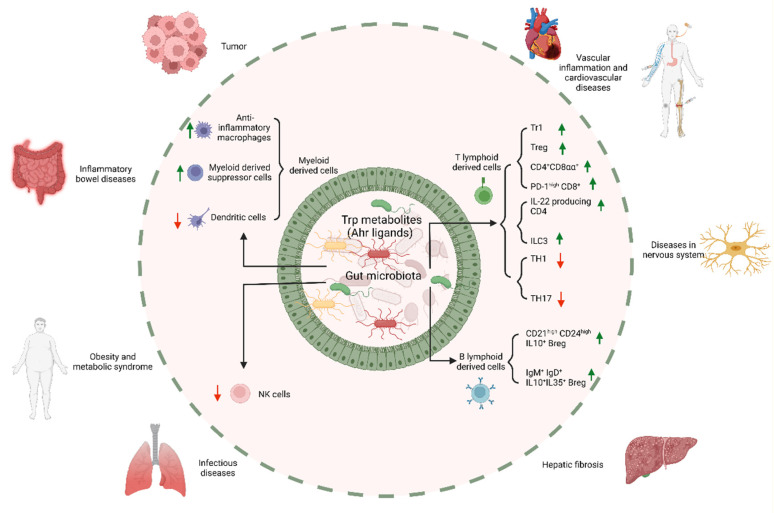
Effects of Trp metabolites derived from the gut microbiota on the immune cells and immune-associated diseases. Trp metabolites promote the differentiation and/or function of immunosuppressive cells such as anti-inflammatory macrophages, myeloid-derived suppressor cells, Tr1, Treg, CD4^+^CD8αα^+^, PD-1^high^CD8^+^, ILC3, IL-22-producing CD4, CD21^high^CD24^high^IL-10^+^Breg and IgM^+^IgD^+^IL-10^+^IL35^+^Breg, whereas the differentiation and/or function of some immune cells such as dendritic cells, TH1, TH17 and NK cells are inhibited. Trp metabolites derived from the gut microbiota can affect the occurrence and development of some diseases through regulating immune cells, such as inflammatory bowel diseases, vascular inflammation and cardiovascular diseases, hepatic fibrosis, diseases in the nervous system, obesity and metabolic syndrome, tumors and infectious diseases.

**Table 1 cells-11-02296-t001:** Gut microbiota species and Trp metabolites.

Metabolite	Bacterial Species		References
Indole	*Achromobacter liquefaciens* *Bacteroides ovatus* *Bacteroides* sp. *Clostridium limosum* *Clostridium* sp. *Corynebacterium acnes* *Citrobacter koser* *Desulfovibrio vulgaris*	*Enterococcus faecalis Escheichia coli* *Proteus vulgaris* *Paracolobactrum coliforme* *Pseudomonas. aeruginosa* *Salmonella enterica* *Vibrio cholerae*	[23,26]
Indole-3-acid-acetic (IAA)	*Bacteroides* spp. *Bacteroides thetaiotaomicron* *Bacteroides eggerthii* *Bacteroides ovatus* *Bacteroides fragilis* *Bifidobacterium adolescentis* *Bifidobacterium longum* subsp. *longum* *Bifidobacterium* spp. *Bifidobacterium pseudolongum* *Clostridium sporogenes* *Clostridium bartlettii* *Clostridium* spp. *Clostridium difficile*	*Clostridium lituseburense* *Clostridium paraputrificum* *Clostridium perfringens* *Clostridium putrefaciens* *Clostridium saccharolyticum* *Clostridium sticklandii* *Clostridium subterminale* *Escherichia coli* *Eubacterium hallii* *Eubacterium cylindroides* *Parabacteroides distasonis* *Peptostreptococcus asscharolyticus*	[5,20,27]
Indole3-lactic acid (ILA)	*Anaerostipes hadrus* *Anaerostipes caccae* *Bacteroides thetaiotaomicron* *Bacteroides eggerthii* *Bacteroides ovatus* *Bacteroides fragilis* *Bifidobacterium adolescentis* *Bifidobacterium bifidum* *Bifidobacterium longum* subsp. *infantis* *Bifidobacterium longum* subsp. *longum* *Bifidobacterium pseudolongum* *Bifidobacterium* spp. *Clostridium bartlettii* *Clostridium perfringens*	*Clostridium sporogenes* *Clostridium saccharolyticum* *Clostridia* *Escherichia. coli* *Eubacterium rectale* *Eubacterium cylindroides* *Faecalibacterium prausnitzii* *Lactobacillus murinus* *Lactobacillus paracasei* *Lactobacillus reuteri* *Megamonas hypermegale* *Parabacteroides distasonis* *Peptostreptococcus asscharolyticus*	[17,23,28,31]
Indole-3-propionic acid (IPA)	*Bacteroides* *Clostridium sporogenes* *Clostridia* *Peptostreptococcus* spp. *Escherichia. coli* *Lactobacillus* *Peptostreptococcus russellii*	*Peptostreptococcus asscharolyticus* *Peptostreptococcus russellii* *Peptostreptococcus anaerobius* *Peptostreptococcus stomatis*	[20,23,27,29,32]
Indoleacrylic acid (IA)	*Clostridium sporogenes* *Peptostreptococcus* spp. *Peptostreptococcus. russellii* *Peptostreptococcus anaerobius* *Peptostreptococcus stomatis*		[27,29,33]
Indole-3-aldehyde (IAld)	*Lactobacillus johnsonii* *Lactobacillus. reuteri* *Lactobacillus. acidophilus* *Lactobacillus. murinus* *Lactobacillus* spp.		[17,34,35]
Tryptamine	*Bacteroides* *Clostridium sporogenes* *Escherichia. coli* *Firmicutes C. sporogenes* *Ruminococcus gnavus*		[30,36]
Indole-3-acetaldehyde (IAAld)	*Escherichia coli*		[37]
3-methylindole (skatole)	*Bacteroides* spp. *Bacteroides thetaiotaomicron* *Butyrivibrio fibrisolvens* *Clostridium bartlettii* *Clostridium* spp. *Clostridium scatologenes*	*Clostridium drakei* *Eubacterium cylindroides* *Eubacterium rectale* *Lactobacillus* spp. *Megamonas hypermegale* *Parabacteroides distasonis*	[5,28,38,39,40]
3-hydroxyanthranilic acid (3-HAA)	*Pseudomonas fluorescens strain KU-7* *Burkholderia cepacia J2315*		[41,42]

## Data Availability

Not applicable.

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
