# Peer review of "Gut Microbiota-Derived Tryptophan Metabolites Maintain Gut and Systemic Homeostasis"

_cells, 2022, doi:10.3390/cells11152296_

Round 1

Reviewer 1 Report

This review by Su X. and colleagues describes the involvement of tryptophan metabolites produced by the intestinal microbiota, and their associated signaling pathways, in gut homeostasis and systemic immunity. The authors highlight the effects of Trp metabolism on immune cells (macrophages and regulatory cells) and link its deregulation to several diseases. Finally, they develop potential applications of Trp metabolites in immunotherapy.

The manuscript is divided into several sections, which are well structured and organized. Sentences follow one another to present the many publications of the literature and their associated data, which sometimes makes reading a bit difficult. However, the figures are informative and well designed, which makes it easier to assimilate all the information.

Comments:

1) In figure 1, color codes could be used to facilitate the visualization of the 2 main pathways: the direct transformation and the Kyn pathway. In general, the quality of the figures could be improved as they come out a bit blurry.

2) A table would be welcome to summarize all bacterial species (presented in §2, lines 65-99) that generate the Trp metabolites and the associated publications.

3) Paragraph 3.2.5 (lines 370-381) mainly presents the anti-tumor effect of Trp metabolites. This paragraph should only present their involvement in cancer (part that is to be developed), and the therapeutic aspect should be placed in paragraph 4.

4) The paragraph 4 (lines 389-406) would benefit from further development. It is very short, and it would be interesting to develop in more detail the different mechanisms described in the data presented, and how they can be used in immunotherapy.

Minor comments:

1) In the abstract lines 15 and 16, abbreviations of the Trp metabolites are not necessary and can be removed (especially since they contain typing errors).

2) In the abstract line 17, please change “system immunity” by “systemic immunity”.

3) The sentence corresponding to lines 96-97 “However, there should have that…” can be improved.

4) The abbreviation “LP” line 182 must be defined.

5) Paragraph 3.1.2 (lines 199-237) should be organized like the previous one with subparagraph numbering: 1). Regulatory CD4+Foxp3+T cells; and 2). Regulatory type 1 (Tr1) cells.

Reviewer 2 Report

The paper is devoted to the analysis of literature data concerning the Trp and its metabolites. In living organisms, the essential amino acid tryptophan plays the role of a key metabolite, as it is a precursor of such important metabolic pathways as serotonin and kynurenine pathways. Changes in the concentration of compounds formed from L-Trp due to insufficient intake of tryptophan or disturbances in its degradation can be associated with many diseases. In this review, the authors tried to analyze the role of gut microbiota- derived Trp in the path physiology, especially in different immune cells. The article discusses how tryptophan metabolites produced by the intestinal microbiota can influence the immune response through the aryl hydrocarbon receptor.

In the abstract of the manuscript, the authors claim to have summarized recent developments in the subject of the proposed manuscript. However, it is difficult to agree with this statement. Keyword searches in the National Library of Medicine database (https://pubmed.ncbi.nlm.nih.gov/) reveal at least twenty more papers in the last two years in this area, but these papers are not discussed in this manuscript. We would like to draw special attention of the authors to the following recent publications:

1.      Yan X , Yan J , Xiang Q , Wang F , Dai H , Huang K , Fang L , Yao H , Wang L , Zhang W . Fructooligosaccharides protect against OVA-induced food allergy in mice by regulating the Th17/Treg cell balance using tryptophan metabolites. Food Funct. 2021 Apr 7;12(7):3191-3205. doi: 10.1039/d0fo03371e. Epub 2021 Mar 18. PMID: 33735338.

2.      Khoshnevisan K, Chehrehgosha M, Conant M, Meftah AM, Baharifar H, Ejtahed HS, Angoorani P, Gholami M, Sharifi F, Maleki H, Larijani B, Khorramizadeh MR. Interactive relationship between Trp metabolites and gut microbiota: The impact on human pathology of disease. J Appl Microbiol. 2022 Jun;132(6):4186-4207. doi: 10.1111/jam.15533. Epub 2022 Mar 29. PMID: 35304801.

3.      Borisova MA, Snytnikova OA, Litvinova EA, Achasova KM, Babochkina TI, Pindyurin AV, Tsentalovich YP, Kozhevnikova EN. Fucose Ameliorates Tryptophan Metabolism and Behavioral Abnormalities in a Mouse Model of Chronic Colitis. Nutrients. 2020 Feb 11;12(2):445. doi: 10.3390/nu12020445. PMID: 32053891; PMCID: PMC7071335.

The second significant remark concerns section 2 and the beginning of section 3. The text of these sections does not contain new information and is very similar to the text of article (Khoshnevisan et al., 2022), even the illustrative material (Figures 1, 2 and 4) is similar to the illustrations of this work. I think that this section should be substantially changed.

Section 3.2.5. Lines 371-372: “We have already made a remarkable progress in understanding of the influence of gut microbiota and/or its metabolites on the onset and progression of tumors [129]”. Ref 129, is this your publication?  Could you describe in more detail the progress made?

Introduction. Lines 50-53. It is necessary to indicate references to the works that are implied.

In general, the review on the problem is insufficient. There is a doubt about a part of the text that this part looks like a plagiarism of the work indicated above (Khoshnevisan et al., 2022). For these reasons I think that the manuscript could not be published in the present form and requires significant improvements.

Round 2

Reviewer 2 Report

The authors of the manuscript took into account the comments of my first review, and changes and additions were made to the text. I am satisfied with the work done. Some English language moderation is required. Now the manuscript is suitable for publication in the journal Cells.